# SpikeForest, reproducible web-facing ground-truth validation of automated neural spike sorters

Jeremy Magland[1]*, James J Jun[1], Elizabeth Lovero[2], Alexander J Morley[3], Cole Lincoln Hurwitz[4], Alessio Paolo Buccino[5], Samuel Garcia[6], Alex H Barnett[1]

[1]Center for Computational Mathematics, Flatiron Institute, New York, United States; [2]Scientific Computing Core, Flatiron Institute, New York, United States; [3]Medical Research Council Brain Network Dynamics Unit, University of Oxford, Oxford, United Kingdom; [4]Institute for Adaptive and Neural Computation Informatics, University of Edinburgh, Edinburgh, United Kingdom; [5]Centre for IntegrativeNeuroplasticity (CINPLA), University of Oslo, Oslo, Norway; [6]Centre de Recherche en Neuroscience de Lyon, Université de Lyon, Lyon, France

**Abstract** Spike sorting is a crucial step in electrophysiological studies of neuronal activity. While many spike sorting packages are available, there is little consensus about which are most accurate under different experimental conditions. SpikeForest is an open-source and reproducible software suite that benchmarks the performance of automated spike sorting algorithms across an extensive, curated database of ground-truth electrophysiological recordings, displaying results interactively on a continuously-updating website. With contributions from eleven laboratories, our database currently comprises 650 recordings (1.3 TB total size) with around 35,000 ground-truth units. These data include paired intracellular/extracellular recordings and state-of-the-art simulated recordings. Ten of the most popular spike sorting codes are wrapped in a Python package and evaluated on a compute cluster using an automated pipeline. SpikeForest documents community progress in automated spike sorting, and guides neuroscientists to an optimal choice of sorter and parameters for a wide range of probes and brain regions.

**\*For correspondence:**
jmagland@flatironinstitute.org

**Competing interests:** The authors declare that no competing interests exist.

## Introduction

### Background

Direct electrical recording of extracellular potentials (*Buzsáki, 2004*; *Seymour et al., 2017*) is one of the most popular modalities for studying neural activity since it is possible to determine, with sub-millisecond time resolution, individual firing events from hundreds (potentially thousands) of cells, and to track the activity of individual neurons over hours or days. Recordings are acquired either from within the living animal (in vivo) or from extracted tissue (ex vivo), at electrodes separated by typically 5–25 μm, with baseline noise on the order of 10 μV RMS and 10–30 kHz sampling rate. Probes for in vivo use—which are usually needle-like to minimize tissue damage during insertion—include microwire monotrodes (*Hubel, 1957*; *Nicolelis et al., 1997*), tetrodes (*Gray et al., 1995*; *Harris et al., 2000*; *Dhawale et al., 2017*), and multi-shank probes (with typically 1–4 columns of electrodes per shank) on silicon (*Csicsvari et al., 2003*; *Buzsáki, 2004*; *Jun et al., 2017b*) or polymer (*Kuo et al., 2013*; *Chung et al., 2019*) substrates. Multiple such probes are often combined into arrays to cover a larger volume in tandem. For ex vivo use (e.g., explanted retina), planar, two-dimensional multi-electrode arrays (MEAs) are common, allowing channel counts of up to tens of

thousands (*Eversmann et al., 2003*; *Litke et al., 2004*; *Berdondini et al., 2005*; *Yuan et al., 2016*; *Tsai et al., 2017*).

Spike sorting is an essential computational step needed to isolate the activity of individual neurons, or units, within extracellular recordings which combine noisy signals from many neurons. Historically, this procedure has relied on manual steps (*Hazan et al., 2006*; *Prentice et al., 2011*; *Rossant et al., 2016*): putative waveforms crossing an amplitude threshold are visualized in a low-dimensional space (either using peak amplitudes or dimensionality reduction techniques), then clusters are separated by eye. While manual spike sorting is manageable with small numbers of recording channels, the rapid growth in channel counts and data volume in recent years as well as the requirement for reproducibility and objectivity demand automated approaches.

Most automated algorithms apply a sequence of steps that include filtering, detection, dimension reduction, and clustering, although these may be combined with (or replaced by) many other approaches such as template matching (*Prentice et al., 2011*; *Pillow et al., 2013*; *Pachitariu et al., 2016*), dictionary learning or basis pursuit (*Carlson et al., 2014*; *Ekanadham et al., 2014*), and independent component analysis (*Takahashi et al., 2002*; *Buccino et al., 2018*). The past 20 years have seen major efforts to improve these algorithms, with recent work focusing on the challenges arising from probe drift (changing spike waveform shapes), spatiotemporally overlapping spikes, and massive data volumes. We will not attempt a full review here, but instead refer the reader to, for example *Fee et al., 1996*; *Lewicki, 1998*; *Quiroga, 2012*; *Einevoll et al., 2012*; *Rey et al., 2015*; *Lefebvre et al., 2016*; *Hennig et al., 2019*; *Carlson and Carin, 2019*.

In the last few years, many automated spike sorters have been released and are in wide use. Yet, there is little consensus about which is the best choice for a given probe, brain region and experiment type. Often, decisions are based not on evidence of accuracy or performance but rather on the ease of installation or usage, or historical precedent. Thus, the goals of extracting the highest quality results from experiments and of improving reproducibility across laboratories (*Denker et al., 2018*; *Harris et al., 2016*) make objective comparison of the available automated spike sorters a pressing concern.

## Prior work

One approach to assessing spike sorter accuracy is to devise intrinsic quality metrics that are applied to each sorted unit, quantifying, for instance, the feature-space isolation of a cluster of firing events (*Pouzat et al., 2002*; *Schmitzer-Torbert et al., 2005*; *Hill et al., 2011*; *Neymotin et al., 2011*; *Barnett et al., 2016*; *Chung et al., 2017*). Another approach is to use biophysical validation methods such as examining cross-correlograms or discovered place fields (*Li et al., 2015*; *Chung et al., 2017*). However, the gold standard, when possible, is to evaluate the sorter by comparing with ground-truth data, that is using recordings where the spike train for one or more units is known a priori. Laboratory acquisition of such recordings is difficult and time-consuming, demanding simultaneous *paired* extracellular and intra-/juxta-cellular probes (*Harris et al., 2000*; *Franke et al., 2015*; *Neto et al., 2016*; *Yger et al., 2018*; *Allen et al., 2018*; *Marques-Smith et al., 2018a*). Since the number of ground-truth units collected in this way is currently small (one per recording), *hybrid* recordings (where known synthetic firing events are added to experimental data) (*Marre et al., 2012*; *Steinmetz, 2015*; *Rossant et al., 2016*; *Pachitariu et al., 2016*; *Wouters et al., 2019*), and biophysically detailed *simulated* recordings (*Camuñas-Mesa and Quiroga, 2013*; *Hagen et al., 2015*; *Gratiy et al., 2018*; *Buccino and Einevoll, 2019*), which can contain 1–2 orders of magnitude more ground-truth units, have also been made available for the purpose of method validation.

Recently, such ground-truth data have been used to compare new spike sorting algorithms against preexisting ones (*Einevoll et al., 2012*; *Pachitariu et al., 2016*; *Chung et al., 2017*; *Jun et al., 2017a*; *Lee et al., 2017*; *Yger et al., 2018*). However, the choice of accuracy metrics, sorters, data sets, parameters, and code versions varies among studies, making few of the results reproducible, transparent, or comprehensive enough to be of long-term use for the community. To alleviate these issues, a small number of groups initiated web-facing projects to benchmark spike sorter accuracy, notably G-Node (*Franke et al., 2012*), a phy hybrid study (*Steinmetz, 2015*) and spikesortingtest (*Mitelut, 2016*). To our knowledge, these unmaintained projects are either small-scale snapshots or are only partially realized. Yet, in the related area of calcium imaging, leaderboard-style comparison efforts have been more useful for establishing community benchmarks (*Freeman, 2015*; *Berens et al., 2018*).

## SpikeForest

We have addressed the above needs by creating and deploying the SpikeForest software suite. SpikeForest comprises a large database of electrophysiological recordings with ground truth (collected from the community), a parallel processing pipeline that benchmarks the performance of many automated spike sorters, and an interactive website that allows for in-depth exploration of the results. At present, the database includes hundreds of recordings, of the types specified above (paired and state-of-the-art biophysical simulation), contributed by eleven laboratories and containing more than 30,000 ground-truth units. Our pipeline runs the various sorters on the recordings, then finds, for each ground-truth unit, the sorted unit whose firing train is the best match, and finally computes metrics involving the numbers of correct, missing, and false positive spikes. A set of accuracy evaluation metrics are then derived per ground-truth unit for each sorter. By averaging results from many units of a similar recording type, we provide high-level accuracy summaries for each sorter in various experimental settings. In order to understand the failure modes of each sorter, SpikeForest further provides various interactive plots.

A central aim of this project is to maximize the transparency and reproducibility of the analyses. To this end, all data—the set of recordings, their ground-truth firings, and firing outputs from all sorters—are available for public download via our Python API. SpikeForest itself is open-source, as are the wrappers to all sorters, the Docker (*Merkel, 2014*) containers, and all of the parameter settings used in the current study results. In fact, code to rerun any sorting task may be requested via the web interface, and is auto-generated on the fly. Both the code and the formulae (for accuracy, SNR, and other metrics) are documented on the site, with links to the source code repositories.

## Contribution

Our work has three main objectives. The primary goal is to aid neuroscientists in selecting the optimal spike sorting software (and algorithm parameters) for their particular probe, brain region, or application. A second goal is to spur improvements in current and future spike sorting software by providing standardized evaluation criteria. This has already begun to happen as developers of some spike sorting algorithms have already made improvements in direct response to this project. As a byproduct, and in collaboration with the SpikeInterface project (*Buccino et al., 2019*), we achieve a third objective of providing a software package which enables laboratories to run a suite of many popular, open-source, automatic spike sorters, on their own recordings via a unified Python interface.

## Results

In conjunction with the SpikeInterface project (*Buccino et al., 2019*), the SpikeForest Python package provides standardized wrappers for the following popular spike sorters: HerdingSpikes2 (*Hilgen et al., 2017*), IronClust (Jun et al., in preparation), JRCLUST (*Jun et al., 2017a*), KiloSort (*Pachitariu et al., 2016*), KiloSort2 (*Pachitariu et al., 2019*), Klusta (*Rossant et al., 2016*), MountainSort4 (*Chung et al., 2017*), SpyKING CIRCUS (*Yger et al., 2018*), Tridesclous (*Garcia and Pouzat, 2019*), and WaveClus (*Chaure et al., 2018*; *Quiroga et al., 2004*). Details of each of these algorithms are provided in *Table 1*. Since each of these spike sorters operates within a unique computing environment, we utilize Docker (*Merkel, 2014*) and Singularity (*Kurtzer et al., 2017*) containers to rigorously encapsulate the versions and prerequisites for each algorithm, ensuring independent verifiability of results, and circumventing software library conflicts.

The electrophysiology recordings (together with ground-truth information) registered in SpikeForest are organized into *studies*, and studies are then grouped into *study sets*. *Table 1* details all study sets presently in the system. Recordings within a study set share a common origin (e.g., laboratory) and type (e.g., paired), whereas recordings within the same study are associated with very similar simulation parameters or experimental conditions.

Each time the collection of spike sorting algorithms and ground-truth datasets are updated, our pipeline, depicted in *Figure 1*, reruns the ten sorters on the recordings. It then finds, for each ground-truth unit, the sorted unit whose firing train is the best match, and finally computes metrics involving the numbers of correct, missing, and false positive spikes. A set of accuracy evaluation metrics are then derived per ground-truth unit for each sorter and displayed on the website.

**Table 1.** Table of spike sorting algorithms currently included in the SpikeForest analysis.

Each algorithm is registered into the system via a Python wrapper. A Docker recipe defines the operating system and environment where the sorter is run. Algorithms with asterisks were updated and optimized using SpikeForest data. For the other algorithms, we used the default or recommended parameters.

| Sorting algorithm | Language | Notes |
| --- | --- | --- |
| HerdingSpikes2* | Python | Designed for large-scale, high-density multielectrode arrays. See *Hilgen et al., 2017*. |
| IronClust* | MATLAB and CUDA | Derived from JRCLUST. See Jun et al., in preparation. |
| JRCLUST | MATLAB and CUDA | Designed for high-density silicon probes. See *Jun et al., 2017a*. |
| KiloSort | MATLAB and CUDA | Template matching. See *Pachitariu et al., 2016*. |
| KiloSort2 | MATLAB and CUDA | Derived from KiloSort. See *Pachitariu et al., 2019*. |
| Klusta | Python | Expectation-Maximization masked clustering. See *Rossant et al., 2016*. |
| MountainSort4 | Python and C++ | Density-based clustering via ISO-SPLIT. See *Chung et al., 2017*. |
| SpyKING CIRCUS* | Python and MPI | Density-based clustering and template matching. See *Yger et al., 2018*. |
| Tridesclous* | Python and OpenCL | See *Garcia and Pouzat, 2019*. |
| WaveClus | MATLAB | Superparamagnetic clustering. See *Chaure et al., 2018*; *Quiroga et al., 2004*. |

## Web interface

The results of the latest SpikeForest analysis may be found at https://spikeforest.flatironinstitute.org and are updated on a regular basis as the ground-truth recordings, sorting algorithms, and sorting parameters are adjusted based on community input. The central element of this web page is the main results matrix (*Figure 2*) which summarizes results for each sorter listed in *Table 1* (using

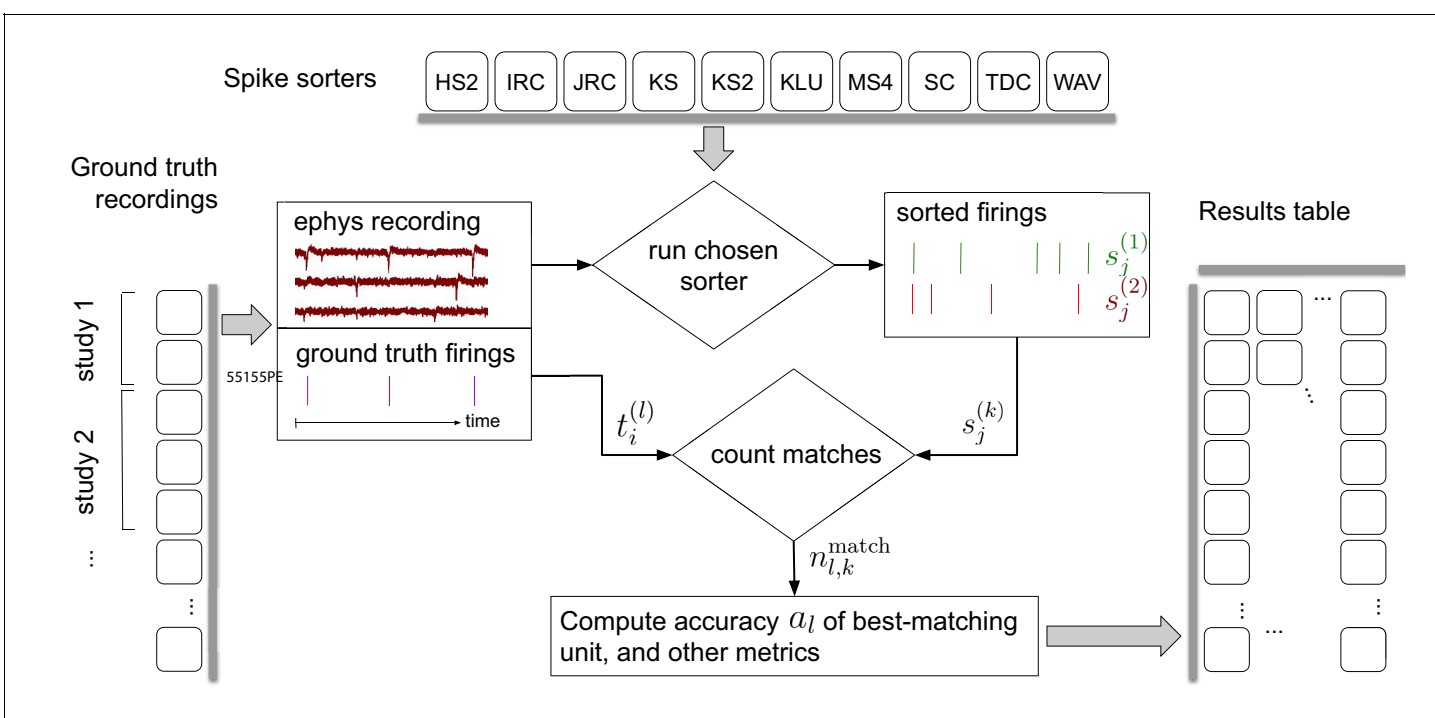

**Figure 1.** Simplified flow diagram of the SpikeForest analysis pipeline. Each in a collection of spike sorting codes (top) are run on each recording with ground truth (left side) to yield a large matrix of sorting results and accuracy metrics (right). See the section on comparison with ground truth for mathematical notations. Recordings are grouped into 'studies', and those into 'study sets'; these share features such as probe type and laboratory of origin. The web interface summarizes the results table by grouping them into study sets (as in *Figure 2*), but also allows drilling down to the single study and recording level. Aspects such as extraction of mean waveforms, representative firing events, and computation of per-unit SNR are not shown, for simplicity.

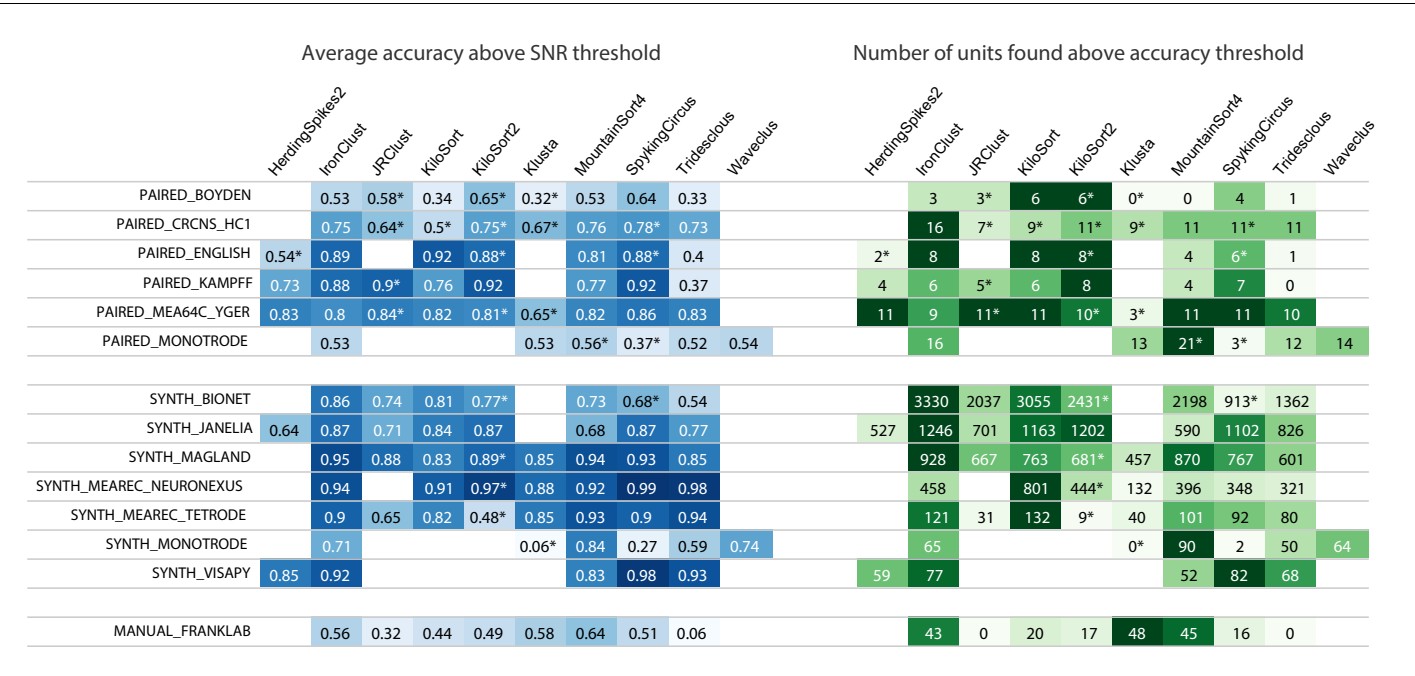

**Figure 2.** Main results table from the SpikeForest website showing aggregated results for 10 algorithms applied to 13 registered study sets. The left columns of the table show the average accuracy (see (5)) obtained from averaging over all ground-truth units with SNR above an adjustable threshold, here set to 8. The right columns show the number of ground-truth units with accuracy above an adjustable threshold, here set to 0.8. The first five study sets contain paired recordings with simultaneous extracellular and juxta- or intra-cellular ground truth acquisitions. The next six contain simulations from various software packages. The SYNTH_JANELIA, obtained from *Pachitariu et al., 2019*, is simulated noise with realistic spike waveforms superimposed at known times. The last study set is a collection of human-curated tetrode data. An asterisk indicates an incomplete (timed out) or failed sorting on a subset of results; in these cases, missing accuracies are imputed using linear regression as described in the Materials and methods. Empty cells correspond to excluded sorter/study set pairs. These results reflect the analysis run of March 23rd, 2020.

formulae defined later by *Equation 5*). The average accuracies are mapped to a color scale (heat map), with darker blue indicating higher accuracy, using a nonlinear mapping designed to highlight differences at the upper end. For the average accuracy table on the left, only ground-truth units with SNR above a user-adjustable threshold are included in the average accuracy calculations; the user may then explore interactively the effect of unit amplitude on the sorting accuracies of all sorters. If a sorter either crashes or times out (>1 hr run time) on any recording in a study set, an asterisk is appended to that accuracy result, and the missing values are imputed using linear regression as described in the Materials and methods section (there is also an option to simply exclude the missing data from the calculation).

The right table of *Figure 2* displays the number of ground truth units with accuracy above a user-adjustable threshold (0.8 by default), regardless of SNR. This latter table may be useful for determining which sorters should be used for applications that benefit from a high yield of accurately sorted units and where the acceptable accuracy threshold is known. The website also allows easy switching between three evaluation metrics (accuracy, precision, and recall) as described in the section on comparison with ground truth.

Clicking on any result expands the row into its breakdown across studies. Further breakdowns are possible by clicking on the study names to reveal individual recordings. Clicking on any result brings up a scatter plot of accuracy vs. SNR for each ground-truth unit for that study/sorter pair (e.g., *Figure 3*, left side). Additional information can then be obtained by clicking on the markers for individual units, revealing individual spike waveforms (e.g., *Figure 3*, right side).

## Parallel operation and run times

Since neuroscientist users also need to compare the efficiencies (speeds) of algorithms, we measure total computation time for each algorithm on each study, and provide this as an option for display

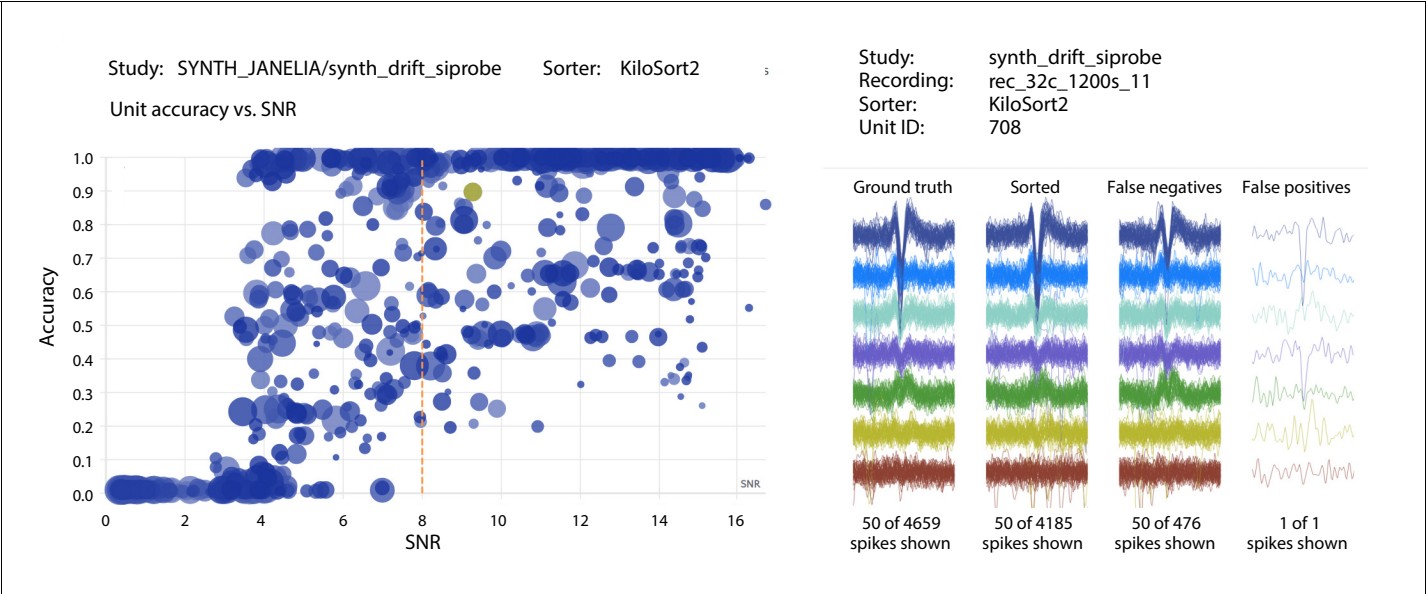

**Figure 3.** Screenshots from the SpikeForest website. (left) Scatter plot of accuracy vs. SNR for each ground-truth unit, for a particular sorter (KiloSort2) and study (a simulated drift dataset from the SYNTH_JANELIA study set). The SNR threshold for the main table calculation is shown as a dashed line, and the user-selected unit is highlighted. Marker area is proportional to the number of events, and the color indicates the particular recording within the study. (right) A subset of spike waveforms (overlaid) corresponding to the selected ground truth unit, in four categories: ground truth, sorted, false negative, and false positive.

on the website via a heat map. Run times are measured using our cluster pipeline, which allocates a single core to each sorting job on shared-memory multi-core machines (with GPU resources as needed). Since many jobs thus share I/O and RAM bandwidth on a given node, these cannot be taken as accurate indicators of speeds in ideal, or even typical, laboratory settings. We emphasize that our pipeline has been optimized for generation and updating of the accuracy results, not for speed benchmarking. For these reasons, we will not present run time comparisons in this paper, referring the interested reader to the website. Here we only note that older sorters such as Klusta can be over 30 times slower than more recent GPU-enabled sorters such as KiloSort and IronClust.

At present, the total compute time for the 650 recordings and 10 sorters is 380 core hours, yet it takes only 3–4 hr (excluding failing jobs) to complete this analysis when run in parallel on our compute cluster with up to 100–200 jobs running simultaneously (typically 14 jobs per node). Since the system automatically detects which results require updating, the pipeline may be run on a daily basis utilizing minimal compute resources for the usual situation where few (if any) updates are needed.

## Sorter accuracy comparison results

We now draw some initial conclusions about the relative performances of the spike sorters based on the threshold choices in *Figure 2*. No single spike sorter emerged as the top performer in all study sets, with IronClust, KiloSort2, MountainSort4, and SpyKING CIRCUS each appearing among the most accurate in at least six of the study sets.

The higher average accuracy of KiloSort2 over its predecessor KiloSort is evident, especially for paired recordings. However, in synthetic studies, particularly tetrodes, KiloSort finds more units above accuracy 0.8 than KiloSort2. Scatter plots (e.g., *Figure 3*, left side) show that KiloSort2 can retain high accuracy down to lower SNR than other sorters, but not for all such low-SNR units. While KiloSort2 was among the best performers for six of the study sets, KiloSort and KiloSort2 had higher numbers of crashes than any of the other sorters, including crashing on every one of the SYNTH_VIS-APY recordings. It is likely that modifications to sorting parameters could reduce the number of crashes, but attempts so far, including contacting the author, have not yet fixed this problem. In the synthetic datasets, KiloSort2 had the largest number of *false positive units* (distinct from the false positive rate of a single unit), but this is not currently reported by SpikeForest (see Discussion).

IronClust appears among the top average accuracies for eight of the study sets, and is especially strong for the simulated and drifting recordings. For most study sets, IronClust has improved accuracy over its predecessor JRCLUST, and is also improved in terms of speed and reliability (no crashes observed). Although a substantial portion of the development of the IronClust software took place while it had access to the SpikeForest ground truth datasets, the same sorting parameters are used across all studies, limiting the potential for overfitting (see Discussion).

MountainSort4 is among the top performers for six of the study sets (based on the average accuracy table) and does particularly well for the low-channel-count datasets (monotrodes and tetrodes). It is not surprising that MountainSort4 is the top performer for MANUAL_FRANKLAB because that data source was used in development of the algorithm (*Chung et al., 2017*).

When considering the left table (average accuracy), SpyKING CIRCUS is among the best sorters for ten study sets. However, it ranks a lot lower in the unit count table on the right of *Figure 2*. This was an example of a sorter that improved over a period of months as a result of using SpikeForest for benchmarking.

HerdingSpikes2 was developed for high-density MEA probes and uses a 2D estimate of the spike location, hence was applied only for recordings with a sufficiently planar electrode array structure (this excluded tetrodes and linear probes). For PAIRED_MEA64C_YGER its performance was similar to other top sorters, but in the other study sets, it was somewhat less accurate. One advantage of HerdingSpikes2 not highlighted in the results table is that it is computationally efficient for large arrays, even without using a GPU.

Tridesclous is among the top performers for both MEAREC study sets and for PAIRED_MEA64-C_YGER, but had a substantially lower accuracy for most of the other datasets. This algorithm appears to struggle with lower-SNR units.

Klusta is substantially less accurate than other sorters in most of the study sets, apart from MANUAL_FRANKLAB where, surprisingly, it found the most units above accuracy 0.8 of any sorter. It also has one of the highest crash/timeout rates.

The version of WaveClus used in SpikeForest is only suited for (and only run on) monotrodes; a new version of WaveClus now supports polytrodes, but we have not yet integrated it. We included both paired and synthetic monotrode study sets with studies taken from selected single electrodes of other recordings. Four sorters (HerdingSpikes2, JRCLUST, KiloSort, and KiloSort2) were unable to sort this type of data. Of those that could, MountainSort4 was the most accurate, with accuracies slightly higher than WaveClus.

An eleventh algorithm, Yet Another Spike Sorter (YASS) (*Lee et al., 2017*), was not included in the comparison because, even after considerable effort and reaching out to the authors, its performance was too poor, leading us to suspect an installation or configuration problem. We plan to include YASS in a future version of the analysis.

## Precision and recall results

Depending on the scientific question being asked, researchers may want to place a greater importance on maximizing either precision or recall. Precision is the complement of the false positives rate which corresponds to spikes incorrectly labeled as coming from some true neuron. A low precision (high number of false positives) may result in illusory correlations between units and a potentially false conclusion that the neurons are interacting, or may result in a false correlation between a unit firing and some stimulus or task. A low recall, on the other hand, means a large fraction of the true firing events are missed, causing a general reduction in putative firing rates, and also possibly introducing false correlations.

*Figure 4* shows aggregated precision and recall scores for the results in the main SpikeForest table, again using the SNR threshold of 8 (keep in mind that conclusions can depend strongly on this threshold). We will not attempt to summarize the entire set of results, only to make two observations. For the paired studies, the sorters that have the highest precisions are IronClust, KiloSort2, MountainSort4, and SpyKING CIRCUS. For the paired and manual studies, precisions are generally a lot lower than recalls, across most sorters. Interestingly, this is *not* generally true for the synthetic studies (where often the precision is higher than recall), indicating that, despite the sophistication of many of these simulations, they may not yet be duplicating the firing and noise statistics of real-world electrophysiology recordings.

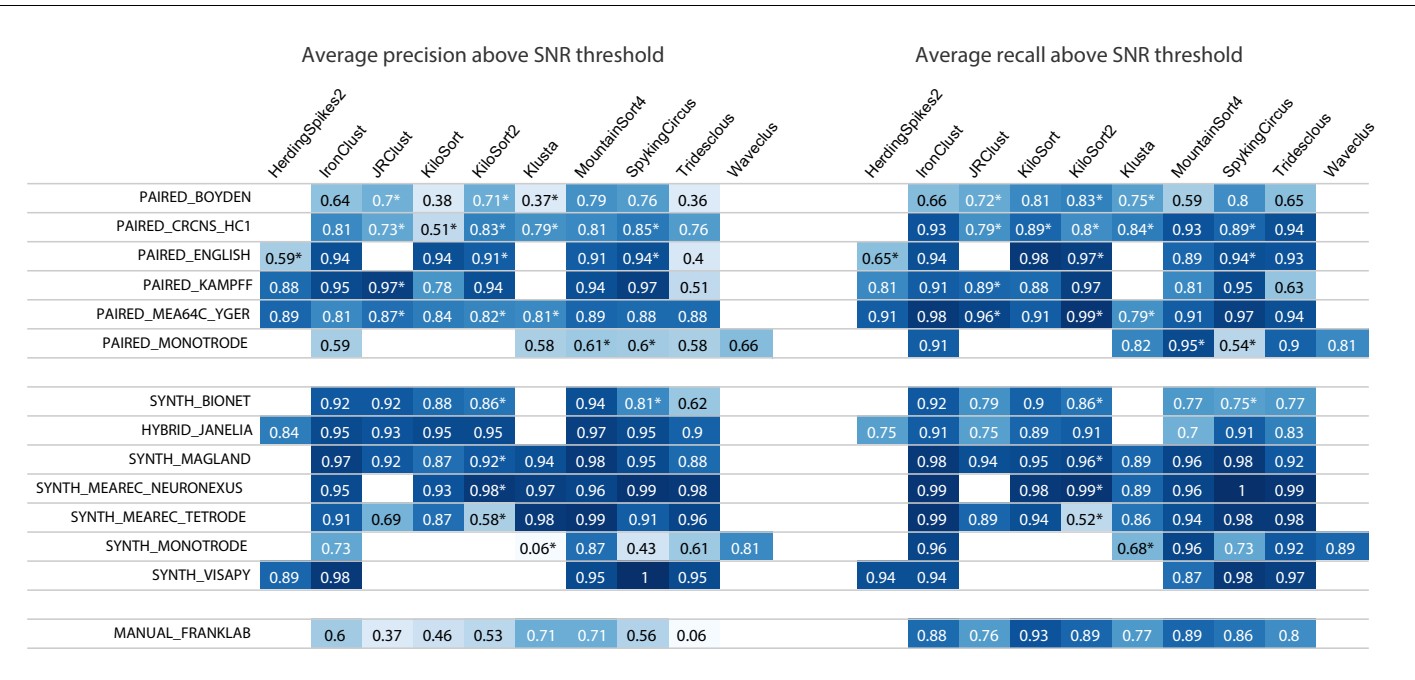

**Figure 4.** Results table from the SpikeForest website, similar to the left side of *Figure 2* except showing aggregated precision and recall scores rather than accuracy. Precision measures how well the algorithm avoids false positives, whereas recall is the complement of the false negative rate. An asterisk indicates an incomplete (timed out) or failed sorting on a subset of results; in these cases, missing accuracies are imputed using linear regression as described in the Materials and methods. Empty cells correspond to excluded sorter/study set pairs. These results reflect the analysis run of March 23rd, 2020.

## How well can quality metrics predict accuracy?

In addition to informing the selection of a spike sorter, our SpikeForest analysis provides an unprecedented opportunity to compare various quality metrics that can be used to accept or reject sorted units when ground truth is not available (i.e., in a laboratory setting). For each sorter, what is the quality metric (or combination thereof) most predictive of actual accuracy? *Figure 5* is based on the SYNTH_JANELIA tetrode study and shows the relationships between ground-truth accuracy and three metrics of the sorted units: SNR, mean firing rate, and inter-spike interval violation rate (ISI-vr) (*Hill et al., 2011*). The latter is the ratio between the number of refractory period violations (2.5 ms threshold) and the expected number of violations under a Poisson spiking assumption. We observe that these relationships are *highly dependent* on the spike sorter. For IronClust, the SNR and log ISI-vr are predictive of accuracy, whereas firing rate is much less predictive. For KiloSort and SpyKING CIRCUS, firing rate and SNR are both predictive, but log ISI-vr does not appear to correlate. For KiloSort2 and MountainSort4, firing rate is the only predictive metric of the three. The final column in this plot shows that a linear combination of metrics is a better predictor than any metric alone. We note that this predictive ability will also depend on the recording type, and, in this case, fidelity of the simulation. For these reasons it is an important future task to extend this type of study across the entire database; see the discussion below.

## Discussion

We have introduced a Python framework and public website for evaluating and comparing many popular spike sorting algorithms by running them on a large and diverse set of curated electrophysiology datasets with ground truth. We have described the principal features of the website. The Materials and methods section details how we prepared ground truth data of varying types (paired, synthetic, and manually-curated) and apply the algorithms in a uniform, transparent, and unbiased manner. We have summarized initial findings comparing sorter accuracies, and invite the reader to explore the continually updating results on the live site.

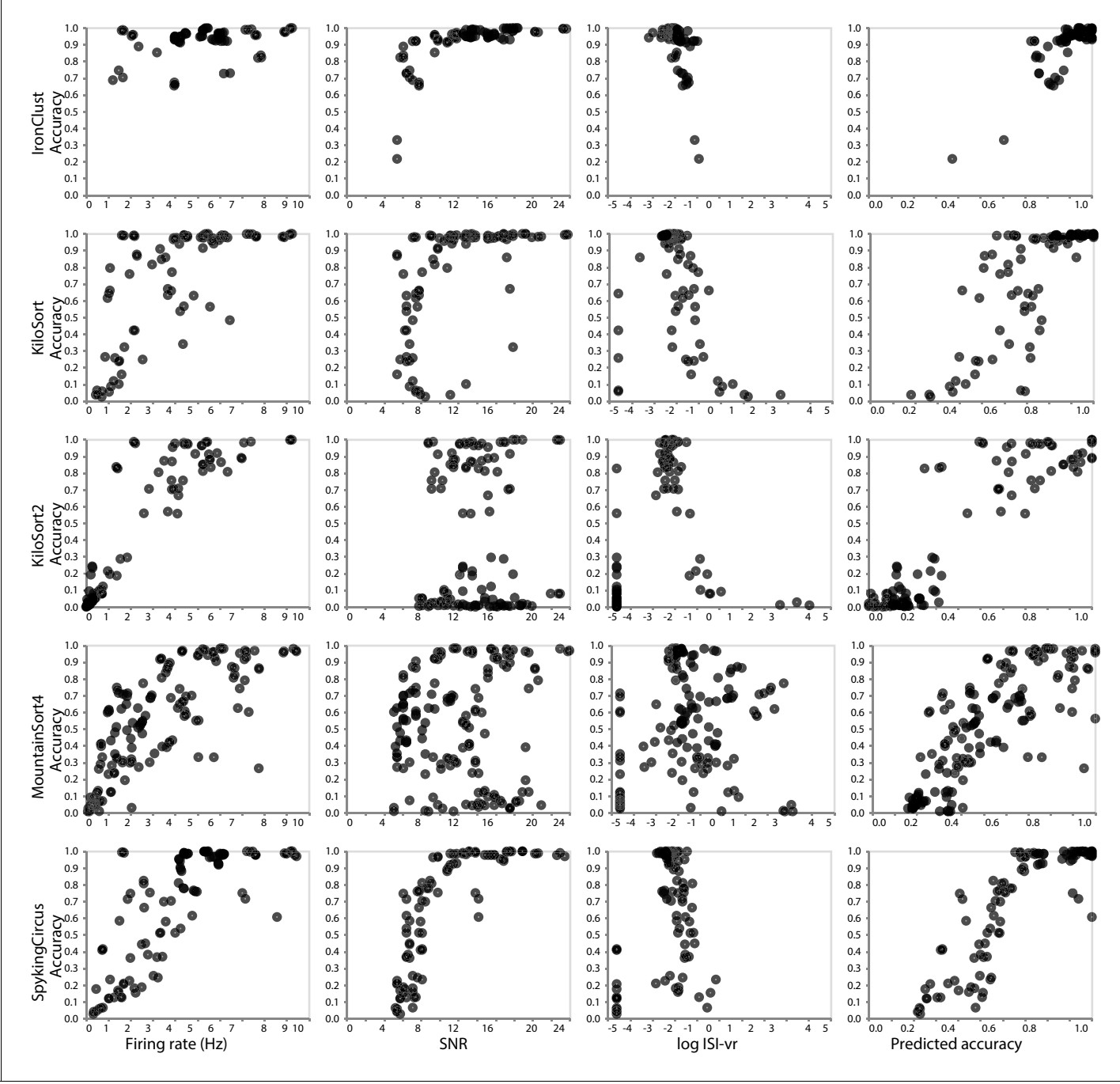

**Figure 5.** Relationship between ground-truth accuracy and three quality metrics for all sorted units (with SNR ≥5), for the SYNTH_JANELIA tetrode study and five spike sorting algorithms. Each marker represents a sorted unit. The x-axis of the plots in the final column is the predicted accuracy via linear regression using all three predictors (SNR, firing rate, and log ISI-vr).

One conclusion is that (as of this time) there is no single sorter that is the most accurate across a diversity of probe types, experimental conditions, metric types and SNR cut-offs. Rather, several different sorters are optimal in different settings. For example, MountainSort4 performs especially well for low channel counts (tetrodes and monotrodes), IronClust excels in the simulated and drifting datasets, while KiloSort2 retains higher accuracy (for units that are found) at lower SNR than other sorters.

Our contribution also helps to overcome some practical issues with neurophysiology research in the laboratory. Traditionally, spike sorting codes bring special requirements for installation and custom input/output file formats, and also require expertise in sorting parameter selection and using visualization and curation utilities intimately tied to that code. This problem has been exacerbated by the use of GPU computing to improve performance in recently developed codes. It is therefore challenging, and unusual, for a laboratory to have more than one or two installed sorters for in-house evaluation. Part of this obstacle is overcome by the SpikeInterface project (*Buccino et al., 2019*), which SpikeForest utilizes, but another part is solved by the Docker and Singularity containers that we have developed to fully capture the operating system environments needed by each sorter.

In the past, comparisons between spike sorting algorithms have been biased or limited. Almost all have been presented in the context of a new or improved method, and so the choice of simulations used for evaluation can often be (unintentionally) biased toward showing that the new method is superior. Given only such reports, it is impractical for readers to verify that the various sorters were used properly and with sensible parameters. The openness and reproducibility of SpikeForest in part remedies this situation.

## Validation approach and challenges

There are many ongoing challenges to the validation of spike sorters. Although human curation is still part of the spike sorting analysis pipeline in most labs, the increase in the potential yield from recently developed high-density recording devices will soon make this step infeasible. We therefore have adopted the philosophy that spike sorting algorithms should be evaluated in an automated reproducible fashion on recordings that we make publicly available, and that, when used for benchmarking purposes, the algorithms should be wrapped and run server-side without the possibility of human curation of their output. This follows evaluation efforts for automated clustering algorithms such as ClustEval (*Wiwie et al., 2015*). We contrast this to 'competition'-style efforts (*Franke et al., 2012*; *Freeman, 2015*; *Berens et al., 2018*) which allow contributions of (potentially non-reproducible) sorting results, and which report accuracy on held-out data whose ground truth is necessarily private, and thus cannot be interrogated by the community.

Although some effort has been made to choose good parameters for each sorter, optimal choices are not guaranteed. We anticipate that a use case for SpikeForest will be finding the best combination of sorter and parameters for a particular recording type. However, automated parameter optimization (as in ClustEval by *Wiwie et al., 2015*) has not yet been implemented, since it would multiply the total CPU cost by a large factor. For now, we encourage the community to contact us with improved settings or algorithms. Indeed, we reached out to the developers of all ten tested sorters in mid-2019 with preliminary results, and several developers (including at least those of SpyKING CIRCUS, HerdingSpikes2, IronClust, and Tridesclous; see *Table 1*) have already used SpikeForest to improve their software. We emphasize that, for reproducibility, sorter versions and parameter choices used for each date-stamped analysis are available in the downloadable analysis archives linked from the website.

This raises the issue of potential *overfitting*. Because all SpikeForest data are public, the community may continue to optimize sorters using SpikeForest as a benchmark, so one might wonder if this will lead to SpikeForest metrics which overestimate true real-world performance. We believe that such bias will be small, and certainly smaller than the bias of studies conducted in order to propose a single sorting algorithm, for the following reasons. 1) Our database is large and diverse, comprising 13 different types of study sets. It is difficult to imagine overfitting to 650 different recordings by optimizing even dozens of parameters, and, while we allow the potential (as yet unused) for different parameter choices for different studies, multiple recordings would still have to be overfit by one parameter set. 2) All code and parameter settings are available for public inspection, making the above style of parameter and algorithm hacking obvious, were it to occur. We feel that the benefit to the community of improved, validated sorters far outweighs the risk of overfitting. Naturally, SpikeForest results *are* biased towards the types of neurons and conditions for which ground-truth data exist; by keeping results for different study sets separate we expose such biases as much as possible. In the long run, we appreciate that *both* held-out and open data benchmarks will play a useful role in comparison and standardization.

Another issue is the paucity of paired ground-truth units in the database, a consequence of the time-consuming nature of their collection. For instance the study set PAIRED_KAMPFF has 15 units,

of which only 11 have sufficient SNR to be sortable by any of the sorters. Therefore a *single* false split or merge by a sorter can lead to variations in reported study-averaged accuracy as large as ±0.05, and dictate the most accurate sorter for that study set. In (small-scale) laboratory pipelines it is possible that such splits or merges would be detected by expert human curation; the point of SpikeForest is to measure the performance of purely automated algorithms. Such variance will be reduced as the size of the ground-truth studies increases.

## Future work

The above issues help inform several specific future goals:

1. We plan to report false positive units. Currently only the one sorted unit which best matches each ground-truth unit is assessed for accuracy. Thus, there is no penalty for a sorter which generates many spurious units that are not present in the data (this is necessarily true for Kilo-Sort in most cases, since the number of returned units is set in advance by the user). Results from such a sorter would of course then be less credible, and, even if examined by an expert, much harder to curate. This failure mode is *not* revealed by paired recordings; however, in simulations (where every single firing is ground truth) a new metric capturing the fraction of such spurious units will be included (see *Buccino et al., 2019*).

2. SpikeForest does not directly address the common laboratory task of deciding, given a new sorting of electrophysiology recordings, which neural units are to be trusted and which discarded. Assessing the credibility of putative neural units output by spike sorters remains a crucial question, and many quality metrics are used in practice, including SNR, firing rate, metrics of cluster isolation or separation from the 'noise cluster', stability, and cross-correlograms (*Pouzat et al., 2002*; *Schmitzer-Torbert et al., 2005*; *Hill et al., 2011*; *Neymotin et al., 2011*; *Barnett et al., 2016*; *Chung et al., 2017*). Yet, because of its scale, the SpikeForest database provides a unique opportunity to tabulate such quality metrics for multiple sorter outputs, then correlate each to the actual ground-truth accuracy, with the goal of assessing the *predictive power* of each metric, or combinations thereof. *Figure 5* showed such a metric comparison for just one of the SpikeForest studies. We are now poised to scale this up to the entire database and a wider set of metrics.

3. Most of the ground truth recordings in our database have a relatively short duration, on the order of 10 min. When waveform drift is not a factor, this duration usually yields enough events per unit to satisfy the requirements of clustering algorithms. However, when waveform drift is present, it is important to assess the performance of spike sorters for a greater range of recording durations. We plan to provide examples of longer duration datasets (on the order of 1–2 hr) in future iterations.

4. It is currently not as easy as it could be to identify common failure modes such as false merges and false splits. We plan to display *confusion matrices* (see, e.g., *Barnett et al., 2016*) between ground-truth and a given sorting; this would also allow comparison between two different sorter outputs.

5. We plan to add accuracy evaluations taylored to specific tasks, such as the ability to handle probe drift and/or long recording durations (currently only three synthetic studies explicitly test drift). The website could, for instance, visualize successful tracking as a function of time.

6. The comparison of CPU time for each sorter is currently sub-optimal, because multiple sorters may be running on one node. We plan to include special timing comparisons on dedicated nodes in order to reflect actual laboratory use cases.

7. The number of sorter job crashes and time-outs needs be reduced, which requires detailed diagnosis on many sorter-recording pairs. For instance, currently KiloSort2 has a larger proportion of crashes or time-outs than many other sorters, yet this is not fully understood and not quantified in accuracy summary results.

8. Finally, we envision that the machinery of SpikeForest could be used for a *web-based* spike sorting platform, to which users would upload their data (which becomes public), run possibly more than one sorter, visualize, curate, and download the results. This could render all spike sorting and human curation affecting the downstream science analysis accessible and reproducible.

We anticipate that, as the use and scale of spike sorting as a tool continues to grow, SpikeForest will become a resource for comparison and validation of sorter codes and encourage more rigorous reproducibility in neuroscience. To this end, we seek contributions and input from the

electrophysiology community, both to optimize parameter settings for existing algorithms, and to further expand the set of algorithms and ground-truth recordings included in the analysis.

## Materials and methods

### Ground-truth recordings

The thirteen study sets included in the SpikeForest analysis are detailed in *Table 2* and are grouped according to type (paired, simulated, curated).

#### Paired recordings

We selected 145 paired recordings from raw extracellular recordings that were publicly released or otherwise provided to us by four laboratories (*Henze et al., 2000*; *Harris et al., 2000*; *Henze et al., 2009*; *Neto et al., 2016*; *Allen et al., 2018*; *Marques-Smith et al., 2018a*; *Marques-Smith et al., 2018b*; *Yger et al., 2018*; *Spampinato et al., 2018*). The intracellular spike times were taken from the author-reported values unless they were not provided (*Henze et al., 2009*). 93 of these we prepared from raw Buzsáki laboratory recordings (*Henze et al., 2000*; *Harris et al., 2000*; *Henze et al., 2009*) with ground truth based on the intracellular traces, after excluding time segments containing artifacts due to movement and current injection. The other 52 were obtained from tetrodes or conventional silicon probes in rat hippocampus (PAIRED_CRCNS_HC1) (*Henze et al., 2000*; *Harris et al., 2000*; *Henze et al., 2009*), high-density silicon probes in mice cortex

**Table 2.** Table of study sets currently included in the SpikeForest analysis.

Study sets fall into three categories: paired, synthetic, and curated. Each study set comprises one or more studies, which in turn comprise multiple recordings acquired or generated under the same conditions.

| Study set | # Rec. / # Elec. / Dur. | Source lab. | Description |
|---|---|---|---|
| Paired intra/extracellular | | | |
| PAIRED_BOYDEN | 19 / 32ch / 6-10min | E. Boyden | Subselected from 64, 128, or 256-ch. probes, mouse cortex |
| PAIRED_CRCNS_HC1 | 93 / 4-6ch / 6-12min | G. Buzsaki | Tetrodes or silicon probe (one shank) in rat hippocampus |
| PAIRED_ENGLISH | 29 / 4-32ch / 1-36min | D. English | Hybrid juxtacellular-Si probe, behaving mouse, various regions |
| PAIRED_KAMPFF | 15 / 32ch / 9-20min | A. Kampff | Subselected from 374, 127, or 32-ch. probes, mouse cortex |
| PAIRED_MEA64C_YGER | 18 / 64ch / 5min | O. Marre | Subselected from 252-ch. MEA, mouse retina |
| PAIRED_MONOTRODE | 100 / 1ch / 5-20min | Boyden, Kampff, Marre, Buzsaki | Subselected from paired recordings from four labs |
| Simulation | | | |
| SYNTH_BIONET | 36 / 60ch / 15min | AIBS | BioNet simulation containing no drift, monotonic drift, and random jumps; used by JRCLUST, IronClust |
| SYNTH_JANELIA | 60 / 4-64ch / 5-20min | M. Pachitariu | Distributed with KiloSort2, with and without simulated drift |
| SYNTH_MAGLAND | 80 / 8ch / 10min | Flatiron Inst. | Synthetic waveforms, Gaussian noise, varying SNR, channel count and unit count |
| SYNTH_MEAREC_NEURONEX | 60 / 32ch / 10min | A. Buccino | Simulated using MEAREC, varying SNR and unit count |
| SYNTH_MEAREC_TETRODE | 40 / 4ch / 10min | A. Buccino | Simulated using MEAREC, varying SNR and unit count |
| SYNTH_MONOTRODE | 111 / 1ch / 10min | Q. Quiroga | Simulated by Quiroga lab by mixing averaged real spike waveforms |
| SYNTH_VISAPY | 6 / 30ch / 5min | G. Einevoll | Generated using VISAPy simulator |
| Human curated | | | |
| MANUAL_FRANKLAB | 21 / 4ch / 10-40min | L. Frank | Three manual curations of the same recordings |

(PAIRED_KAMPFF and PAIRED_BOYDEN) (*Neto et al., 2016*; *Allen et al., 2018*; *Marques-Smith et al., 2018a*), and high-density MEAs in mice retina (PAIRED_MEA64C_YGER) (*Yger et al., 2018*; *Spampinato et al., 2018*). We also include 29 new paired recordings in awake behaving mice (PAIRED_ENGLISH), from a hybrid probe comprising a juxtacellular electrode glued on top of a Neuronexus silicon probe (similar to *English et al., 2017* but with only ~20 µm separation). These come from the hippocampus, neocortex and thalamus. We generated a monotrode version (PAIRED_MONOTRODE) of various paired recordings by randomly sampling one channel from each recording session to uniformly span the SNR range (25 units per study).

## Synthetic recordings

*Table 1* provides details on the ten algorithms included in the SpikeForest analysis. To overcome the limited number of units offered by the paired ground truth, we added simulated ground-truth study sets that were independently generated by five laboratories (*Camuñas-Mesa and Quiroga, 2013*; *Hagen et al., 2015*; *Chung et al., 2017*; *Gratiy et al., 2018*; *Buccino and Einevoll, 2019*). The simulators vary in their biophysical details, computational speeds, and configurable parameters. Simulations based on phenomenological models tend to be fast and easily configurable (e.g., SYNTH_MAGLAND, identical to the simulations in *Chung et al., 2017* except with iid Gaussian noise), while biophysical simulators such as SYNTH_VISAPY (*Hagen et al., 2015*) and SYNTH_BIONET (*Gratiy et al., 2018*) use synaptically connected, morphologically detailed neurons to achieve high fidelity at the expense of computational speed. SYNTH_MONOTRODE (*Camuñas-Mesa and Quiroga, 2013*) and SYNTH_MEAREC (*Buccino and Einevoll, 2019*) take an intermediate approach by generating spike waveform templates based on single-neuron simulations and randomly placing the spike waveforms conforming to pre-specified ISI distributions.

SYNTH_BIONET was generated from the Allen Institute's BioNet simulator (*Gratiy et al., 2018*; *Jun et al., 2017a*) running on the computing resources provided by the Neuroscience Gateway (*Sivagnanam et al., 2013*). We simulated a column of synaptically connected neurons ($n = 708$, $200 \times 200 \times 600 \, \mu\mathrm{m}^3$) based on the rat cortical NEURON models (*Hines and Carnevale, 1997*; *Ascoli et al., 2007*; *Mitelut, 2017*; *Jun et al., 2017a*) by capturing the spike waveforms at four vertical columns of densely-spaced electrodes (2 µm vertical, 16 µm horizontal pitch, 600 channels). Linear probe drift was simulated by subsampling the electrodes to match the Neuropixels site layout (20–25 µm pitch) and vertically shifting the electrode positions as a function of time. To achieve smooth linear motion, a 2D-interpolation based on a Gaussian kernel was applied (0.5 µm vertical spacing, 16 µm total displacement for 16 min). Based on the linear probe drift simulation, we generated a 'shuffled' version to mimic the fast probe displacements during animal movement by subdividing the recordings into 32 time segments and randomly shuffling (30 s per segment).

The simulation entitled SYNTH_JANELIA was developed by J. Colonell (*Pachitariu et al., 2019*) and uses average unit waveforms from a densely spaced electrode array (5 µm pitch, 15 × 17 layout) collected by the *Kampff, 2018* laboratory. The waveform templates were inserted at randomly selected channels and time points after being multiplied by an amplitude-scaling factor drawn from a Gamma distribution. The baseline noise was randomly generated to match the power spectrum observed from a Neuropixels recording, and a spatiotemporal smoothing was applied to induce correlation between nearby channels and time samples. The original simulation generated either static or sinusoidal drift (±10 µm displacement, 600 s period) with 1200s duration (30 KS/s) on the Neuropixels layout (64 channels, 25 µm pitch, staggered four columns). We trimmed the original simulations to generate shorter recording durations (600, 1200s) and channel counts (4, 16, 32, 64 channels) to study the effects of these parameters on sorting accuracy.

## Manually sorted recordings

Based on the requests of the electrophysiology community, we included a single study set of *manually* curated tetrode sortings (MANUAL_FRANKLAB) (*Chung et al., 2017*); we emphasize that, for this study set only, *there is no ground-truth data*. Accuracies are reported relative to three independent manual sortings of the same tetrode recording session in a chronically implanted rat hippocampus. We subdivided the entire recording duration to generate three different durations (600, 1200, 2400 s) to study the effect of duration on sorting accuracy, but did not find a consistent relationship across all sorters.

## Registration

A content-addressable storage database is used to store the file content of each recording, and all data is available for public download via our Python API. Since these files are indexed according to their SHA-1 hashes, their content is guaranteed not to have changed even when the mechanism for retrieving the data evolves over time, ensuring long-term repeatability of the analysis.

## Sorting algorithms and wrappers

*Table 2* provides details on the ten algorithms included in the SpikeForest analysis. For each spike sorter, SpikeForest contains a Python wrapper and a Docker (and Singularity) container defining the exact execution environment including all necessary dependencies. For the sorters that are implemented in MATLAB, the MATLAB Compiler tool was used to generate standalone binary files inserted into the containers so that a MATLAB license is not required to run spike sorting. The Spike-Forest framework flexibly allows running of each sorter either within the native operating system or within the container. The former method is useful during development or in an environment where the spike sorting software is already installed. The latter is crucial for ensuring reproducibility and for avoiding conflicts between sorters due to incompatible dependencies. The Python wrappers make use of the spikesorters package of the SpikeInterface project (*Buccino et al., 2019*) for passing the parameters and executing the sorter. All sorters operate on the raw (unfiltered) recordings.

An eleventh algorithm, YASS, was incorporated into the Python package, but was not included in the comparison, as discussed in the Results section.

## Comparison with ground truth

Depending on the experimental context, false negatives (missed events) and false positives may have different relative importance for the researcher. Thus the SpikeForest website allows switching between three evaluation metrics for the comparison with ground truth: precision (which penalizes only false positives), recall (which penalizes only false negatives), and an overall accuracy metric (which balances the two). For each sorter-recording pair these are computed by comparing the output of spike sorting (spike times and labels) with the ground-truth timings associated with the recording, as follows.

We first consider a sorted unit $k$ and a ground-truth unit $l$, and describe how the events in the spike train for $k$ are matched to ground-truth events of $l$. Let $s_1^{(k)}, \ldots, s_{M_k}^{(k)}$ be the spike train (time-stamps) associated with sorted unit $k$, and let $t_1^{(l)}, \ldots, t_{N_l}^{(l)}$ be the spike train for ground-truth unit $l$ (see *Figure 1*). Let $\Delta$ be an acceptable firing time error, which we assume is shorter than half the refractory period of any true neuron. We set $\Delta$ to one millisecond; the results are rather insensitive to its exact value. The number of matches of sorted unit $k$ to the ground-truth unit $l$ is

$$n_{l,k}^{\mathrm{match}} := \#\{i : |t_i^{(l)} - s_j^{(k)}| \leq \Delta \text{ for some } j\}. \tag{1}$$

Note that even if more than one sorted event falls within $\pm\Delta$ of a true event, at most one is counted in this matching. The reverse situation—more than one ground-truth event from the same neuron matching to a given sorted event—cannot happen by our assumption about the refractory period. The number of missed events and false positives are then, respectively,

$$n_{l,k}^{\mathrm{miss}} := N_l - n_{l,k}^{\mathrm{match}} , \qquad\qquad n_{l,k}^{\mathrm{fp}} := M_k - n_{l,k}^{\mathrm{match}} , \tag{2}$$

where $N_l$ is the total number of firings of ground-truth unit $l$, and $M_k$ is the total number of found events of sorted unit $k$. Following *Jun et al., 2017b* we define the *accuracy* for this pair as

$$a_{l,k} := \frac{n_{l,k}^{\mathrm{match}}}{n_{l,k}^{\mathrm{match}} + n_{l,k}^{\mathrm{miss}} + n_{l,k}^{\mathrm{fp}}} . \tag{3}$$

Note that this definition of accuracy is a balance between precision and recall, and is similar, but not identical, to the $F_1$ metric (*Zaki and Meira Jr, 2014*, Eq. (17.1)) used to evaluate clustering methods.

Fixing the ground-truth unit $l$, we define its *best matching sorted unit* $\hat{k}_l$ as the sorted unit $k$ with highest accuracy,

$$\hat{k}_l := \arg\max_k a_{l,k} . \tag{4}$$

Now restricting to this best match, we define the accuracy for ground-truth unit $l$ by

$$a_l := a_{l,\hat{k}_l} , \tag{5}$$

and the corresponding precision and recall for this unit by

$$p_l := \frac{n_{l,\hat{k}_l}^{\mathrm{match}}}{n_{l,\hat{k}_l}^{\mathrm{match}} + n_{l,\hat{k}_l}^{\mathrm{fp}}} \qquad\qquad r_l := \frac{n_{l,\hat{k}_l}^{\mathrm{match}}}{n_{l,\hat{k}_l}^{\mathrm{match}} + n_{l,\hat{k}_l}^{\mathrm{miss}}} . \tag{6}$$

Averages of these metrics are then computed for all units $l$ within each study, without weighting by their numbers of events (i.e., treating all units equally). Note that, in the case of a recording with more than one ground-truth unit, it is possible that more than one such unit could share a common best-matching sorted unit, but this could only happen if these ground-truth units had extremely correlated events or if the sorting was highly inaccurate.

The spike sorted units that are considered in the computation of these metrics are only those that are matched to ground-truth units. Therefore, the results shown in the main table do not account for false positive units, that is units found by the spike sorters that are not present in the recording.

## Compensating for missing data

As described above, when a sorting run fails (either crashes or times out), an asterisk is appended to the corresponding table cell, and the average accuracy is calculated based on imputed data using linear regression. There is also an option to simply exclude the missing values, but the problem with this method (which we have encountered) is that sometimes an algorithm will happen to crash on recordings with relatively difficult units, resulting in artificially elevated scores. Imputing by zero has the problem of yielding deceptively low values. We have thus opted to use linear regression to fill in the missing data using values predicted based on the accuracies of other sorters. Specifically, for a given sorter with missing data and a given study set (or study, depending on the level of aggregation), a linear model is estimated for fitting the non-missing values based on the values of all sorters with no missing data. That model is then applied to estimate the missing data. To give some intuition, if a sorter typically has somewhat lower scores than the other sorters and crashes on one recording, then for the purpose of computing average accuracy, the accuracy for that recording will be filled in with a value that is also somewhat lower than the other sorters.

In contrast, when reporting total numbers of units found above an accuracy cut-off (e.g., **Figure 2**, right side), we do not impute, but simply sum the number of non-missing units.

## Signal-to-noise ratio per unit

We define SNR as a property of a single neural unit, either a ground-truth unit or a unit as output by a sorter. It is reported on the website and is used as a cut-off for selection of a subset of ground-truth units for computing average accuracy. SNR is computed on the bandpass-filtered timeseries data.

We first describe our filter used to compute SNR (noting that this is distinct from various filters used internally by the spike sorters). This filter is a bandpass from $f_{\min} = 300$ to $f_{\max} = 6000$, in Hz units. It is applied to each channel by taking the FFT, multiplying by the real-valued frequency filter function

$$A(f) = \frac{1}{2}\sqrt{1 + \mathrm{erf}((f - f_{\min})/100)}\sqrt{1 - \mathrm{erf}((f - f_{\max})/1000)} , \tag{7}$$

where erf is the error function, then taking the inverse FFT. Here, the parameters 100 and 1000 control the roll-off widths in Hz at the low and high ends respectively. From this filtered timeseries and the set of firing times of a unit, the average spike waveform is extracted. SNR is then defined as the ratio between the peak absolute amplitude of this average spike waveform and the estimated noise

on the channel where this peak amplitude occurs. The noise is estimated as the median absolute deviation of the filtered timeseries data divided by 0.6745, which gives a robust estimate of the standard deviation of the noise (*Quiroga et al., 2004*, Sec. 3.1).

## Analysis pipeline

The analysis pipeline of SpikeForest depicted in *Figure 6* is built using Python utilities developed by the authors for creating shareable and reproducible scientific workflows. This system provides a formal method for creating well-defined Python procedures that operate on input parameters and files and produce output files. These are known as *processors*. SpikeForest defines processors for running the spike sorters, computing properties of ground-truth units (e.g., SNR), comparing the spike sorting outputs with ground truth, and computing summary data for the plots shown on the website. Once these processors are set up, the framework provides several advantages including: (a) automatic execution of processing inside Singularity or Docker containers; (b) automatic caching of processing results; and (c) job management and queueing mechanisms for running batches of processing on a compute cluster. This allows the analysis pipeline to be defined as a standard Python script, with a simple nested loop iterating through the sorters and ground-truth datasets (as in *Figure 1*). The script may then be configured to run in a variety of settings: on a standard workstation for development, testing, or reproducing of a subset of results; a shared computer with large memory and many cores; or on a compute cluster.

A crucial and novel feature of our framework is that all files (both input and output) are represented by SHA-1 URIs, for example of the form: `sha1://88b62db6fc467b83ba0693453c59c5-f538e20d5c/firings_true.mda`.

The hexadecimal code embedded in the URI is the SHA-1 hash of the content of the file, and therefore this URI uniquely identifies the desired file (in this case the ground-truth firing data for one of the SpikeForest recordings) without specifying the actual location of the data. In contrast, explicit location references (e.g., a path on the local computer, an IP address, or a web URL) can be problematic because over time data archives may stop being maintained, may change locations, or files may be renamed or updated with new content. The SHA-1 URI system alleviates these difficulties by separating the mechanism for storing archives of files from the representation of these files, via universal hash strings within scripts.

All SpikeForest input recordings, ground-truth data, sorting outputs, and other processing data are stored in a public content-addressable storage (CAS) system called *kachery*. The API of a kachery database simply allows downloading of files referenced only by their SHA-1 hashes (or URIs).

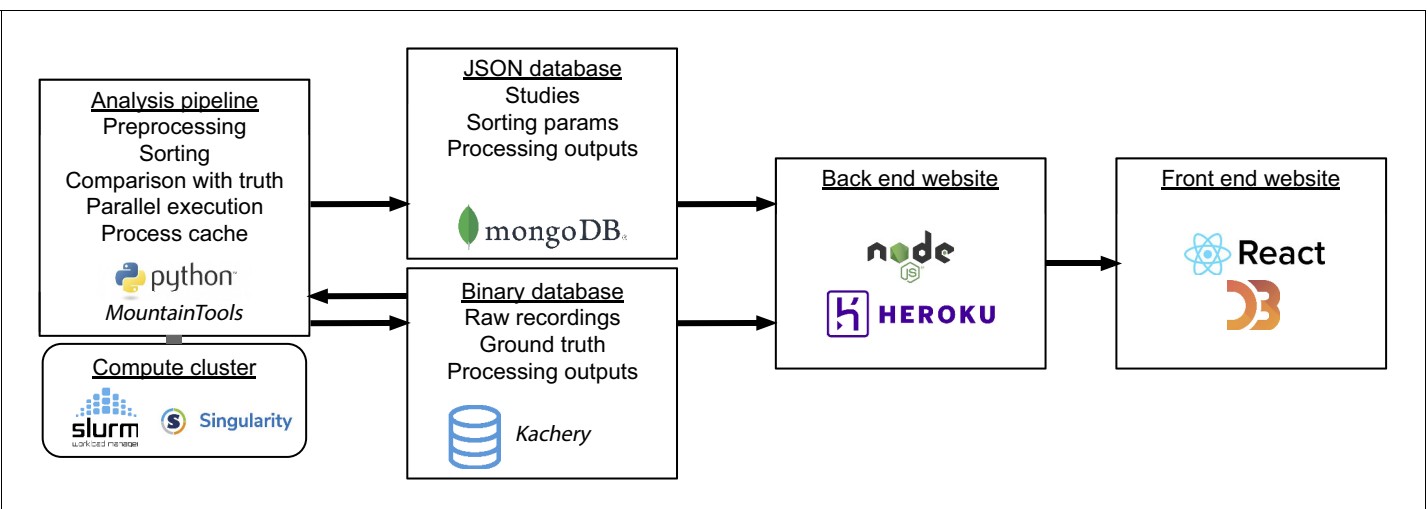

**Figure 6.** Interaction of software and hardware components of the SpikeForest system, showing the flow of data from the server-side analysis (left) to the user's web browser (right). The processing pipeline automatically detects which sorting jobs need to be updated and runs these in parallel as needed on a compute cluster. Processing results are uploaded to two databases, one for relatively small JSON files and the other for large binary content. A NodeJS application pulls data from these databases in order to show the most up-to-date results on the front-end website.

Therefore, as long as the environment is configured to point to a kachery database with the relevant files available, the SpikeForest pipeline may be executed (in whole or in part) on any computer connected to the internet.

The automatic caching capability we developed is also crucial for the SpikeForest system. If updates are made to either the database of ground-truth recordings or to the sorting algorithms and parameters, the system can automatically detect which processing needs to be rerun. In this way, the pipeline can simply be executed in full at regular intervals, and the website will continuously update with the latest changes. This facilitates a conceptually simple method for adding new datasets or sorting algorithms because changes are represented by modifications to pipeline configuration files.

A limitation of the reliance on caching is the implicit assumption that the sorting algorithms are deterministic, that is given the same inputs, parameters, and code, the outputs should be exactly the same. However, this assumption can be violated by unfixed random seeds or even differences between implementations of floating-point arithmetic on different hardware. Controlling these factors is currently out of scope for SpikeForest, but we aim to measure the degree of stability (*Barnett et al., 2016*) with respect to repeated runs of each sorting task in future versions.

To ensure that all results may be reproduced at a later date and/or independently verified by a third party, the Python wrapper for each spike sorter includes a reference to a Docker image containing the entire operating system and environment where the sorter is installed. Our system handles the automatic download of these images (if not already on the local computer) and the execution of the sorting inside Docker or Singularity containers. For development and testing purposes, it is also possible to configure the pipeline to run outside the container, provided that the sorter software is installed on the operating system.

The output of processing is a single JSON file (only around 15 MB at this time) containing SHA-1 URI references to all recording files, ground-truth data, containers, spike sorting parameters, sorting outputs, comparisons with ground truth, and other information used by the website. It also contains meta-information about the sorting runs such as the execution times and console outputs. This JSON file is itself uploaded to the kachery database and is represented by a SHA-1 URI accessible on the website. The archive section of the website contains references to these files for all past analyses, allowing tracking of sorter performance over time.

Finally, the data from this JSON output file is loaded into a MongoDB database for efficient access by the website's front end.

## Website front end

The primary user interface for the SpikeForest platform is an isomorphic JavaScript web application with overall structure as shown on the right side of *Figure 6*. Built from reusable React components, the front end utilizes the D3 library to render the interactive tables and plots. A Node.JS backend, organized using the Redux state container, queries a MongoDB database to retrieve JSON files for each datatype and generate the comparative visualizations. For more detailed plots, like the spike sprays, larger data objects are retrieved via the content-addressable storage database.

We optimized the interface for efficient hierarchical navigation through the results and rapid loading and interactive response (for instance, when a user adjusts the SNR or accuracy cut-off slider bar). When the user clicks on results at the individual recording or unit level, they are taken to a page (with an auto-generated, shareable URL) specific to the study in question. This page allows one-click comparison of the sorters on this study. Clicking on individual units in the scatter plot for a given sorter-study pair brings up spike waveforms for that unit (*Figure 3*) and a link to a sub-page with details specific to the particular sorter-recording run. This latter page includes sorter parameters used, the console output of the run and a link to an auto-generated Python script with human-readable documentation to reproduce that sorter run within SpikeForest. A permanent top-level menu bar allows access to all meta-data about sorters (as in *Table 1*), study sets (as in *Table 2*, also allowing hierarchical drilling down to the individual recording level), historical snapshots, metrics used, and an explanation of the project. All of the website data and results sub-pages are automatically generated from the SpikeForest databases.

### Enabling users to run all spike sorters locally on their own data

The benefit of having wrappers and Docker images for these ten spike sorting algorithms extends beyond usage within the SpikeForest pipeline. Researchers may also utilize these wrappers and images to spike sort their own data without needing to install and configure the individual sorting packages. As mentioned in the previous subsection, the website provides, for each sorting result, a self-contained Python script for reproducing that result offline. This script may easily be modified to operate on datasets that are not registered within the SpikeForest database. Here is an excerpt of the auto-generated script for running SpyKING CIRCUS on a dataset:

```
# Run the spike sorting
with hither.config(container='docker://magland/sf-spykingcircus:0.9.2', gpu=-
False):
  sorting_result=sorter.run(
    recording_path=recording_path,
    sorting_out=hither.File(),
    **params
  )
assert sorting_result.success
sorting_path=sorting_result.outputs.sorting_out
```

This is compatible with all three OS platforms: Linux, MacOS and Windows (although, currently, the last of these cannot use GPU-based sorters). Here, hither is a utility within SpikeForest; note that the Docker container and version (located on Docker Hub) is specified in the configuration line. Because we use SpikeInterface, it is easy to switch between a variety of different file formats for the input and output data. The SpikeForest website provides further instructions and examples.

## Acknowledgements

We have benefited from discussions with Leslie Greengard, Eftychios Pnevmatikakis, Andrea Giovannucci, Karel Svoboda, Loren Frank, and the developers of several of the sorting codes, and from the comments of the anonymous reviewers. The BioNet simulations were made possible by Catalin Mitelut and the Neuroscience Gateway Portal. We thank Dan English and colleagues for making their hybrid paired recordings available. We are grateful to the organizers of the Janelia spike sorting workshop, held in March 2018, which helped to inspire this project. The Flatiron Institute is a division of the Simons Foundation.

## Additional information

### Funding
No external funding was received for this work.

### Author contributions
Jeremy Magland, Conceptualization, Data curation, Software, Investigation, Visualization, Methodology, Writing - original draft, Writing - review and editing; James J Jun, Conceptualization, Data curation, Software, Investigation, Methodology, Writing - original draft, Writing - review and editing; Elizabeth Lovero, Software, Investigation, Visualization, Methodology, Writing - original draft, Writing - review and editing; Alexander J Morley, Software, Methodology, Writing - review and editing; Cole Lincoln Hurwitz, Alessio Paolo Buccino, Samuel Garcia, Software, Writing - review and editing; Alex H Barnett, Conceptualization, Supervision, Investigation, Methodology, Writing - original draft, Project administration, Writing - review and editing

### Author ORCIDs
Jeremy Magland https://orcid.org/0000-0002-5286-4375
Elizabeth Lovero http://orcid.org/0000-0002-2642-603X

Alexander J Morley (ID) http://orcid.org/0000-0002-4997-4063
Cole Lincoln Hurwitz (ID) http://orcid.org/0000-0002-2023-1653

**Decision letter and Author response**
Decision letter https://doi.org/10.7554/eLife.55167.sa1
Author response https://doi.org/10.7554/eLife.55167.sa2

## Additional files

### Supplementary files
• Transparent reporting form

### Data availability
All data, including 1.3 TB of raw data and all result files, are publicly available through the SpikeForest API.

The following previously published datasets were used:

| Author(s) | Year | Dataset title | Dataset URL | Database and Identifier |
|---|---|---|---|---|
| Henze DA, Borhegyi Z, Csicsvari J, Mamiya A, Harris D, Buzsáki G | 2009 | Simultaneous intracellular and extracellular recordings from hippocampus region CA1 of anesthetized rats | https://doi.org/10.6080/K02Z13FP | Collaborative Research in Computational Neuroscience, 10.6080/K02Z13FP |
| Spampinato LBG, Esposito E, Yger P, Duebel J, Picaud S, Marre O | 2018 | Ground truth recordings for validation of spike sorting algorithms | https://doi.org/10.5281/zenodo.1205233 | Zenodo, 10.5281/zenodo.1205233 |
| Jason EC, Jeremy FM, Alex HB, Vanessa MT, Angela CT, Kye YL, Kedar GS, Sarah HF, Loren MF, Leslie FG | 2017 | A Fully Automated Approach to Spike Sorting | https://doi.org/10.17632/kmmndvycx8.1 | Mendeley Data, 10.17632/kmmndvycx8.1 |

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
