## [Decision Letter]

**Acceptance summary:**

This paper introduces an online web site focused on evaluating different methods of spike sorting: the computational process that extracts the spiking of single neurons from a raw electrical recording of brain signals. The authors set themselves three goals: to aid scientists in selecting a spike sorter; to spur improvements to sorting algorithms; and to provide easy-to-use software packages. In an area that has been crying out for standardization this web site promises to be an invaluable resource.

**Decision letter after peer review:**

Thank you for submitting your article "SpikeForest: reproducible web-facing ground-truth validation of automated neural spike sorters" for consideration by *eLife*. Your article has been reviewed by three peer reviewers including Markus Meister as the Reviewing Editor and Reviewer #1, and the evaluation has been overseen by a Reviewing Editor and Ronald Calabrese as the Senior Editor.

The reviewers have discussed the reviews with one another and the Reviewing Editor has drafted this decision to help you prepare a revised submission.

Summary:

This paper introduces an online resource focused on evaluating different methods of spike sorting. Spike sorting is an indispensable step in analyzing extracellularly recorded neuronal data, and there have been large advances in the software that performs this task over the past few years. The result is a large number of algorithms and no general consensus as to which of these algorithms works best in each circumstance. Add to that the large number of different recording configurations and brain regions, and the result is substantial uncertainty about which sorter to use in any given situation. The SpikeForest website and comparison framework presented here represents a major step toward addressing that problem. The site maintains data from many extracellular recordings that include ground truth spike times of single units. And it implements about a dozen different spike sorting algorithms that can be applied to the same data sets. The results are analyzed by various quality metrics and compared across spike sorters and data sets in tabular form. The site also offers a dive into the results down to the waveforms of individual sorted units. It is live and extensible both with regards to new data sets and new sorters.

The authors set themselves three goals: to aid scientists in selecting a spike sorter; to spur improvements to sorting algorithms; and to provide easy-to-use software packages. Based on experience in our research groups and some polling of some colleagues, these goals have been achieved already. The paper is very well written and offers an indispensable complement to the website. Here is the consensus review:

Essential revisions:

The paper offers little guidance for how to interpret the output of each sorter. Each of these sorters will produce a number of clusters which do not correspond to single units. Some set of quality metrics must be applied when the user decides how to interpret the sorter's output, and the authors have the opportunity to provide some guidance on that critical topic. This is mentioned as the first future direction, and we would encourage the authors to make a first effort at this in the current manuscript.

The Introduction is quite long and reads more like a review article. Please distil this section down to the key facts that a reader needs to know and move other material to Results section or Discussion section. The first figure, for example, would be better placed at the beginning of the Results section.

All of the ground truth datasets are quite short (~10 min), but standard recordings from neuronal populations can span hours. It seems likely that the number of spikes from a given ground truth unit and the number of spikes from other units has an effect on sorting accuracy. Can this impact be quantified for each sorter? Please comment on whether the results will generalize to longer data sets.

Figure 2: Given that there is space for figures, it would be helpful to show not just the overall accuracy metric, but also the false positive and negatives for each sorter. Of course, this is available on the website, but readers may want this information right away. The caption should explain the meaning of empty cells in the table.

Please add some guidance for why one might choose one performance metric versus another. False positives, for example, can be particularly problematic when assessing correlations between units, while false negatives lead to underestimates of firing rates.

The process whereby a user might take advantage of the containerized sorters could be explained more clearly. This is not the main thrust of the manuscript, but creation of these standard software environments greatly furthers the goal of reproducibility.

Results section, Discussion section and elsewhere: Which of the spike sorters have been optimized on these specific test data? This is mentioned in context of IronClust and SpyKING CIRCUS. Maybe spell out in Table 2 for each sorter how the parameters were chosen and whether they were adapted to the test sets.

---

## [Author Response]

Essential revisions:The paper offers little guidance for how to interpret the output of each sorter. Each of these sorters will produce a number of clusters which do not correspond to single units. Some set of quality metrics must be applied when the user decides how to interpret the sorter's output, and the authors have the opportunity to provide some guidance on that critical topic. This is mentioned as the first future direction, and we would encourage the authors to make a first effort at this in the current manuscript.

We agree. This encouragement led to us performing a preliminary offline analysis that uses one of the SpikeForest study sets to predict accuracy on the basis of three quality metrics of sorted units: SNR, firing rate, and inter-spike-interval violation rate (ISI-vr). We chose these metrics as a starting point since they are commonly used, unambiguous, and straightforward to compute. There are other metrics used in the field that require, e.g. access to the clustering feature space, making their extraction purely from the sorted spike trains much more complicated.

A new subsection entitled "How well can quality metrics predict accuracy?" and a new figure (Figure 6) has been added to the Results section showing some of the results of this analysis. We decided to focus on the SYNTH_JANELIA study set because it contains many units with varying degrees of accuracies. As seen in the new Figure 6 in the revised manuscript, the relationship between accuracy and these three metrics is highly dependent on the spike sorting algorithm used (it also depends to some extent on the type of recording). For IronClust, the SNR and ISI-vr are predictive of accuracy, whereas firing rate is much less predictive. For KiloSort and SpykingCircus, firing rate and SNR are both predictive, but ISI-vr does not appear to correlate strongly. For KiloSort2 and MountainSort4, firing rate is the only predictive metric of the three.

There is a subtle but crucial issue that prevents the immediate expansion of such a correlation analysis to other recording types, as follows. The analysis can only be performed using simulated recordings because there all of the true units are known. For the real (paired) recordings the ground truth timing information is available for only one of the units. Say that we wish to show that a certain per-unit metric value is a good predictor of real-world inaccuracy. We can collect this metric for all sorted units, but we cannot prove that any sorted unit is inaccurate. For instance, while there may be no sorted units that match the ground-truth unit well, each may be an accurate sorting of some other legitimate unit. If the reviewers have any solution to this issue, we would appreciate any suggestions as we move to the next phase.

Here is another way to explain the issue: our current website displays accuracies for each ground truth unit by reporting the extent of matching with the most similar sorted unit. In contrast, the proposed metric exploration requires reporting the accuracy for each sorted unit relative to the best matching ground truth. Therefore, while each marker in the website scatterplot corresponds to a ground truth unit, each marker in the new plots (Figure 6) corresponds to a sorted unit.

Adding such metric comparisons to the website interface, addressing the above issue, and expanding the set of metrics, is a larger undertaking that we expect to start shortly, as we indicate in the future work section.

The Introduction is quite long and reads more like a review article. Please distil this section down to the key facts that a reader needs to know and move other material to Results section or Discussion section. The first figure, for example, would be better placed at the beginning of the Results section.

In response to this suggestion, we have improved the readability by breaking the introduction into subsections and shortening it by three paragraphs (one moved to Results section, one moved to Discussion section, and one removed entirely). The first subsection "Background" and comprises four paragraphs that the reader may skim over if they are already familiar with the field. The next subsection "Prior work" containing two paragraphs that we consider crucial for presenting this work within a larger context of validation efforts. The third subsection, "SpikeForest", gives a high-level overview of the approach presented in the paper. We moved two of the more technical paragraphs from this part of the writeup to Results section and Discussion section and removed one other paragraph entirely. Finally, we have subsection "Contribution" that has the paragraph describing the three main objectives of our work. All references to figures have been removed from the Introduction so that, as suggested by the reviewers, the first figure appears as part of the Results section.

All of the ground truth datasets are quite short (~10 min), but standard recordings from neuronal populations can span hours. It seems likely that the number of spikes from a given ground truth unit and the number of spikes from other units has an effect on sorting accuracy. Can this impact be quantified for each sorter? Please comment on whether the results will generalize to longer data sets.

A new 'future direction' item has been added to the Discussion section addressing our plans to add longer duration recordings with drifting spike waveforms.

We selected relatively short datasets partly because of compute time considerations. Multiple algorithms are run on every dataset and some of these sorters are very slow for longer recordings. We also believe that, when waveform drift is not a factor, around 10 minutes usually yields enough events per unit to satisfy typical clustering algorithms (even a 1 Hz average firing rate gives 600 events over 10 minutes).

To support this claim (for the reviewers but not in the paper), we simulated some additional recordings with varying durations (300, 600, 1200, 2400, and 4800 seconds). Technical details of how these were generated are provided below. Author response image 1 shows that for these simulations, accuracy remains steady with increasing duration across all of the sorters.

**Author response image 1. sa2fig1:** Accuracy vs. duration for nine sorters applied to a study set of simulated recordings.

Another issue associated with longer recordings is waveform drift, which varies quite a bit depending on the probe type and experimental setup. To give an indication of how the different sorters perform in the presence of this type of drift over longer recordings, we regenerated the recordings used above, except adding simulated spike waveform drift. In this case (Author response image 2) we see that there is generally a falloff in accuracy with increasing duration, except for three sorters which appear to be more robust with respect to drift: IronClust, KiloSort, and KiloSort2.

**Author response image 2. sa2fig2:** Same as Author response image 1, except that the simulations included synthetic drift.

Technical details on how these simulations were generated: We modified the Kilosort2 eMouse simulator developed by J. Colonell and M. Pachitariu. This simulator uses averaged unit waveforms from the recordings taken from the Kampff laboratory using a densely spaced electrode array (15 x 17 layout spanning 100 x 102 micrometers). In the drift simulation case, a linear probe motion was generated by uniformly translating a 64-channel probe (Neuropixels layout) by 20 micrometers over 80 min. To study the effect of time duration and channel count on the sorting accuracy, we extracted 16 channels from the original output (64 channels, 80 minutes) by taking contiguous neighboring channels at various time durations (5, 10, 20, 40, 80 min) starting at t=0. Ten recordings were sampled from each channel count and time duration by uniformly varying the channel offsets. The simulation inserted waveform templates at random channels and time points after multiplying them by a random scaling factor drawn from a Gamma distribution. The baseline noise was randomly generated to match the power spectrum observed from a Neuropixels recording, and a spatiotemporal smoothing was applied to induce a correlation between nearby channels and time samples.

Figure 2: Given that there is space for figures, it would be helpful to show not just the overall accuracy metric, but also the false positive and negatives for each sorter. Of course, this is available on the website, but readers may want this information right away. The caption should explain the meaning of empty cells in the table.

A new figure (Figure 5) has been added showing aggregated precision and recall for the main results. A new subsection summarizes and motivates this figure, at the end of the Results section. This led to the interesting observation that generally precision is lower than recall for paired recordings, but more often higher than recall for synthetic, showing that the simulations do not yet capture all of the statistics of real-world recordings.

We have also now added an explanation of empty cells to the captions of the two main result figures.

Please add some guidance for why one might choose one performance metric versus another. False positives, for example, can be particularly problematic when assessing correlations between units, while false negatives lead to underestimates of firing rates.

We have added a paragraph to the Results section, which explains when one might prefer to consider precision or recall over accuracy. The paragraph also refers to the new Figure 5 in the previous response.

The process whereby a user might take advantage of the containerized sorters could be explained more clearly. This is not the main thrust of the manuscript, but creation of these standard software environments greatly furthers the goal of reproducibility.

We have added a new subsection "Running spike sorting on a local machine" which explains this.

Results section, Discussion section and elsewhere: Which of the spike sorters have been optimized on these specific test data? This is mentioned in context of IronClust and SpyKING CIRCUS. Maybe spell out in Table 2 for each sorter how the parameters were chosen and whether they were adapted to the test sets.

Asterisks were added to four of the algorithms in the table, and the following was added to the caption:

"Algorithms with asterisks were updated and optimized using SpikeForest data. For the other algorithms, we used the default or recommended parameters."